# Retrieving Decadal Climate Change from Satellite Radiance Observations—A 100-year CO_2_ Doubling OSSE Demonstration

**DOI:** 10.3390/s20051247

**Published:** 2020-02-25

**Authors:** William L. Smith, Elisabeth Weisz, Robert Knuteson, Henry Revercomb, Daniel Feldman

**Affiliations:** 1Space Science and Engineering Center (SSEC), University of Wisconsin, Madison, WI 53706, USA; elisabeth.weisz@ssec.wisc.edu (E.W.); bob.knuteson@ssec.wisc.edu (R.K.); hankr@ssec.wisc.edu (H.R.); 2Lawrence Berkeley National Laboratory, Berkeley, CA 94720, USA; drfeldman@lbl.gov

**Keywords:** hyperspectral, retrieval, climate change, radiative transfer

## Abstract

Preparing for climate change depends on the observation and prediction of decadal trends of the environmental variables, which have a direct impact on the sustainability of resources affecting the quality of life on our planet. The NASA Climate Absolute Radiance and Refractivity Observatory (CLARREO) mission is proposed to provide climate quality benchmark spectral radiance observations for the purpose of determining the decadal trends of climate variables, and validating and improving the long-range climate model forecasts needed to prepare for the changing climate of the Earth. The CLARREO will serve as an in-orbit, absolute, radiometric standard for the cross-calibration of hyperspectral radiance spectra observed by the international system of polar operational satellite sounding sensors. Here, we demonstrate that the resulting accurately cross-calibrated polar satellite global infrared spectral radiance trends (e.g., from the Metop IASI instrument considered here) can be interpreted in terms of temperature and water vapor profile trends. This demonstration is performed using atmospheric state data generated for a 100-year period from 2000–2099, produced by a numerical climate model prediction that was forced by the doubling of the global average atmospheric CO_2_ over the 100-year period. The vertical profiles and spatial distribution of temperature decadal trends were successfully diagnosed by applying a linear regression profile retrieval algorithm to the simulated hyperspectral radiance spectra for the 100-year period. These results indicate that it is possible to detect decadal trends in atmospheric climate variables from high accuracy all-sky satellite infrared radiance spectra using the linear regression retrieval technique.

## 1. Introduction

Climate change is a reality that must be dealt with for the protection of life, property and sustainable resources for generations to come. Preparing for climate change depends on the observation and prediction of the decadal trends of the environmental variables, such as temperature, moisture and clouds, which have a direct impact on the sustainability of resources affecting the quality of life on our planet. Satellites are an ideal platform for the measurement of climate trends, since they can observe the globe on a time continuous basis. Using radiometric instruments on the international system of polar orbiting and geostationary satellites can in principle provide a set of observations which can be averaged to produce climate variables without a diurnal sampling bias.

However, instrument-dependent radiometric calibration errors and cloud-induced effects on the measured radiances can induce errors in estimating any temperature and moisture trends inferred from satellite radiance measurements. Such errors lead to disagreements between the satellite indirect measurement of thermodynamic variables with direct observations using conventional in-situ surface and upper air balloon devices that lack the global unbiased sampling potential of the international system of Earth orbiting satellites. For example, Spencer and Christy [1], Wentz and Schabel [2], Christy et al. [3] and Mears and Wentz [4], discuss potential problems with atmospheric temperature trends inferred from the early satellite microwave sounding instrument brightness temperature measurements (e.g., Spencer and Christy [5]). Recent comparisons between the University of Alabama at Huntsville (UAH) data atmospheric temperature trend estimates, from satellite microwave brightness temperature measurements, with estimates provided by Remote Sensing Systems (RSSs [6]) analysis of satellite microwave data, show reasonable agreement, although differences still exist (Christy et al. [7]). The RSS 4-decade (1980 to 2020) mean indicates that the total tropospheric temperature trend estimate is about 0.21 K/decade (RSS 2020). However, satellite microwave sounder measurements have a relatively poor vertical resolution that acts to reduce their sensitivity to the lower troposphere, the region most important for man’s livelihood. It is shown here that new hyperspectral infrared satellite measurements (Smith et al. [8]) have the potential for providing the vertical resolution needed to resolve the vertical profile of atmospheric temperature and moisture trends, as needed to more accurately assess the impact of climate change on human life.

The NASA Climate Absolute Radiance and Refractivity Observatory (CLARREO) mission (Wielicki et al. [9]) is proposed to provide climate quality benchmark hyperspectral (i.e., 1000s of spectral channels) radiance observations for the purpose of determining the decadal trends of climate variables, and validating and improving the long-range climate model forecasts needed to prepare for the changing climate of the Earth. The CLARREO mission is designed to have an exceptional absolute accuracy (i.e., < 0.1 K 3-sigma brightness temperature at scene temperature). This performance is consistent with the calibration requirements for detecting the trends of atmospheric temperature and moisture profiles from hyperspectral satellite measurements identified by Liu et al. [10]. 

CLARREO is essentially an International Radiometric Standard in space. The new technologies needed to make the highly accurate observations of infrared spectra for CLARREO have been developed and tested (Best, et al., [11]; Taylor et al., [12,13]; Gero et al., [14]). As described by Wielicki et al. [9], the CLARREO Infrared spectrometer is designed to have a spectral range from 200–2000 cm^−1^, with an unapodized spectral resolution of 0.5 cm^−1^. The instrument’s nadir Field of View will enable a ground resolution of 25–100 km. This spectral and spatial resolution will enable CLARREO to be used in-orbit to cross calibrate operational atmospheric sounding interferometer spectrometers (Tobin, et al. [15]) (e.g., JPSS CrIS, Metop IASI, and the FY-3 HIRAS). The IASI instrument is chosen to be a representative operational satellite sounding instrument for this study to determine if decadal temperature and moisture trends can be determined from operational satellite sounding spectrometers, which have been cross-calibrated to CLARREO’s 0.1 K three sigma absolute accuracy.

The question that the current work addresses is this: based on the CLARREO instrument specification, can unbiased temperature and humidity trends be obtained from retrievals using operational polar satellite hyperspectral infrared radiances (e.g., from the Metop IASI sensor), which have high absolute accuracy through their cross-calibration to a CLARREO radiometric standard in space? This question of whether a practical retrieval technique can deliver sufficiently unbiased, long-term trends under all sky conditions is distinctly different from the question of fundamental information content addressed by previous studies, using fingerprinting approaches (e.g., Liu et al. [10]). Here, this question is addressed by leveraging the well-established Dual Regression (DR) retrieval algorithm (Smith, et al. [16]), coupled to observational simulations using the Climate Community System Model version 3 (CCSM3) output from a 100-year period (2000–2099) climate model prediction, that was forced by the doubling of the global average atmospheric CO_2_ over the 100-year period (Feldman et. al. [17,18,19]. 

The same retrieval methodology has been applied to real hyperspectral observations (Smith, N. et al. [20]) to investigate the potential of hyperspectral satellite measurements to detect climate trends. However, in the real satellite data case it is difficult to make a definitive assessment of climate trend measurement capability, since the available satellite measurement time record is short, and the “truth” is unknown. Here, the underlying climate variable trends from the climate model are known, so it is a matter of retrieving an unbiased estimation of these trends from simulated, quasi-continuous and global radiance spectra provided by the CLARREO cross-calibrated operational polar satellite system. The high absolute accuracy of the radiometric data is required to minimize instrument-dependent spatial and temporal variations in the climate variable retrievals, which would otherwise cause errors in the climate trend retrievals. It is fully recognized that these results based on simulated radiances are more controlled than would be the analysis of observed radiance data. However, in the absence of such data, this investigation can, through the use of a climate model that is necessarily simplistic relative to the actual Earth system, still provide an upper-bound on the value of a CLARREO cross-calibrated operational polar satellite system for diagnosing climate variable trends.

This paper presents a “retrieve and average” approach (Kato et al. [21]) to infer climate variable trends from a long time-series of accurate high spectral resolution satellite infrared radiance measurements. In this approach, atmospheric profiles are retrieved from satellite radiance measurements, and then averaged over climate spatial and temporal scales. As pointed out by Kato et al. [21], earlier approaches (e.g., Leroy et al. [22] and Huang et al. [23]) used temporally, and spatially-averaged, spectral radiance time changes to infer atmospheric and cloud property time changes. Kato et al. [21] improved upon the earlier studies that used model-derived monthly mean atmospheric properties by using a linear regression to retrieve temperature, humidity and cloud property changes from spectral radiance changes. In this sense the Kato approach is similar to the linear regression retrieval approach presented here, except for the handling of clouds, which produce nonlinear relations between radiance and the atmospheric state parameters under variable cloud height conditions.

This paper is organized as follows: First we describe the observing system simulation experiment (OSSE) Radiance Simulation; second, we describe the radiative transfer method used; third, we describe the retrieval method; fourth, we describe the regression training and treatment of clouds; and finally, we present conclusions and a brief discussion.

## 2. OSSE Radiance Simulation

Data from the Climate Community System Model (CCSM3) 100-year carbon dioxide (CO_2_) doubling Observation System Simulation Experiment (OSSE) were used to simulate the European operational satellite Metop Infrared Atmospheric Sounding Interferometer (IASI) Instrument (Hilton et al. [24]), which produces hyperspectral radiance spectra with CLARREO-like information content. The CLARREO OSSE, described in Feldman et al. [17,18,19], uses the CCSM3 model. This follows similar work by Huang et al. [23,25,26,27]. Since the proposed CLARREO infrared sensor will be used to cross-calibrate the international fleet of polar operational sounders in orbit, the conclusions are directly relevant to future observing systems. For this study, IASI is assumed to be a representative operational polar operational satellite instrument, which would be cross-calibrated using the CLARREO to an absolute accuracy of 0.1 K 3-sigma. As stated earlier, the goal of this simulation study was to determine how well decadal trends could be retrieved from hyperspectral infrared radiance observations using the linear regression retrieval technique. Data included monthly mean atmospheric temperature and water vapor profiles and cloud and surface parameters for a ~1.5° grid from the CCSM3. The Principal Component Radiative Transfer Model (PCRTM) (Liu et al. [28]) was used to produce radiance spectra (Section 3, below) using the CCSM3 surface skin temperature, cloud variables, and atmospheric temperature and atmospheric constituent profiles as input, together with satellite measurements of surface emissivity (Zhou et al. [29]).

## 3. Radiative Transfer Method

The Principle Component Radiative Transfer Model (PCRTM), developed for hyperspectral radiance applications by Liu et al. [28,30], has been chosen for computing the clear-sky and cloudy-sky radiance spectra corresponding to any particular surface and atmospheric state, as needed to define the regression retrieval relations used for the retrieval process. It has been shown that PCRTM brightness temperature calculations agree very closely (i.e., <0.1 K) with the Line-by-Line Radiative Transfer Model (LBLRTM) calculations and satellite observations (Saunders et al. [31]). PCRTM is extremely fast, a major asset for handling long-term climate data sets. Two types of regression training sets are used in this study: a dependent sample and an independent sample of surface/cloud/atmospheric profile conditions. The “dependent” sample is a subset of surface, cloud and atmospheric profiles forecast by the climate model, producing the 100-year CO_2_ Observation System Simulation Experiment (OSSE), together with the corresponding radiance spectra calculated from these surface and atmospheric state conditions. The “independent” sample is a contemporary weather-related surface, cloud and atmospheric profile condition training data set based on the SeeBor training database (Borbas et al. [32]). Since the SeeBor training data depicts contemporary weather-related surface, cloud and atmospheric profile conditions, it is completely unrelated to the climate change forecast conditions used to synthesize the spectral radiance observations used for this climate variable retrieval study. Further details of the radiance simulation and retrieval procedures are provided below.

## 4. Retrieval Method

Because top-of-atmosphere infrared spectral radiance measurements are sensitive to the vertical structure of atmospheric temperature and moisture, changes in radiance over time, at least in principle, reflect changes in those underlying atmospheric climate variables. However, retrieving atmospheric variables from satellite radiance observations with the quality required for climate studies is an extremely demanding task. The accuracy of the retrieved variables depends on the accuracy of the radiometric data from which they are inferred, the accuracy of the retrieval algorithm used to transform the radiometric data into atmospheric variables, the accuracy of the forward radiative transfer model used within the retrieval algorithm for the inverse solution of the atmospheric variables from the radiance measurements, and the proper modeling of the influence of the Earth’s surface, clouds, aerosols and minor constituents, which must be accounted for in the radiance observations for the retrieval of the temperature and water vapor profiles to be analyzed in terms of decadal trends. In addition, unless all of these uncertainties are random in time (i.e., the radiances measurements used possess a very high absolute accuracy), there is substantial risk that significant biases can be introduced through instrument-dependent time variations in the radiometric data that yield spurious trends in the retrieved atmospheric variables.

The retrieval problem for clear skies has been shown to be sufficiently linear, that any time and space variations of the retrievals accurately reflect the time and space variations of the radiance measurements. However, clouds significantly complicate the retrieval problem because of nonlinear effects dependent on cloud height. The Dual Regression (DR) surface, cloud and atmospheric variable retrieval algorithm developed by Smith et al. [16] was designed to overcome this complication. The DR approach treats clouds in a manner that retains the approximate linearity between the radiance variations being produced by the climate variables (e.g., atmospheric temperature and humidity), and the climate variable variations being retrieved. It treats clouds by classifying the “cloudy-sky“ linear regression relations by cloud height, and by including the same number of clear radiance spectra/atmospheric profile combinations as cloudy radiance spectra/atmospheric profile combinations within the statistical sample of the data used to calculate the regression coefficients for each cloud height category. The all-sky radiance is a linear combination of opaque cloudy-sky radiance and clear-sky radiance, the cloud fraction being the proportionality factor in this linear relationship.

Thus, the cloudy sky regression relations for each cloud height category can represent all potential cloud fractions, ranging from zero, a clear Field-of-View (FOV), to unity, an opaque cloud overcast FOV. The final all-sky retrieval is a combination of a “clear-sky” regression profile retrieval (based on a training sample of atmospheric profiles and “clear-sky’ radiance spectra), together with the “cloudy-sky” regression retrieval, the reason that the algorithm is named “Dual Regression (DR)”. The “clear-sky” regression retrieval is used as the final “all-sky” solution above the highest cloud level, whereas the “cloudy-sky” regression retrieval is used as the final retrieval below the highest cloud level. The cloud height needed to select the proper regression relations between the observed radiance spectrum and the atmospheric profile is defined as that level (i.e., divergent level) where the “clear-sky” regression retrieval begins to diverge systematically from the “cloudy-sky” regression retrieval below the mean cloud height of the classification being used. The correct cloud height is chosen from the results obtained for all cloud height classifications as that divergent altitude which agrees best with the mean cloud height for the cloud classification from which it was produced. The magnitude of the difference between the “clear-sky” surface skin temperature retrieval and the “cloudy-sky” surface skin temperature retrieval represents the degree of radiance attenuation by cloud. As a result, this difference is used to quality control the below cloud level retrieval output. The retrieval below the cloud top level is considered to be missing when this retrieved surface skin temperature difference is greater than 15 K, or when the regression retrieved optical depth is greater than unity (i.e., the cloud produced radiance attenuation is greater than 63%).

One complication of using the CCSM3 atmospheric state data for climate trend satellite retrieval studies, is that the output is the monthly averages of the atmospheric state variables with a 1.5-degree horizontal resolution. As a result, it is necessary to assume that the atmospheric state profiles are statistically uncorrelated with the surface and cloud conditions when producing retrievals of monthly average atmospheric profiles using PCRTM radiance spectra computed from the model output monthly average atmospheric profiles and their associated cloud and surface skin temperature conditions.

## 5. Regression Training and Treatment of Clouds

Dual Regression (DR) retrievals were performed from monthly average CCSM3 grid point “clear-sky” and “all-sky” simulated radiances. The DR regression training (i.e., regression coefficient generation) was performed using both “dependent” (i.e., climate model profile) and “independent” (i.e., contemporary weather dependent surface and vertical profile observation) data sets. Profiles for 36 randomly selected model grid points per month during the 100-year OSSE period (43,200 grid point soundings) were used for the generation of the “dependent sample” regression training. The weather-related atmospheric profile data set (15,794 clear profiles and 19,948 cloudy sky profiles) described by Smith et al. [16] was used for the “independent sample” regression training. CO_2_ concentration was assumed to vary randomly with a uniform distribution ranging between 370 and 800 ppm, such that the simulated radiance variability covered the full range of the 100-year CO_2_ variation assumed for the climate model OSSE. A randomly chosen noise value, from a normal distribution, with a standard deviation equivalent to a brightness temperature noise of 0.1 K, was assumed (i.e., added to the calculated radiances), in order to condition the radiance covariance matrix for inversion as needed to compute the linear regression coefficients. This added radiance noise also accounts for the spectrally random error produced by the PCRTM and spectrally random noise in the radiance observations.

The effects of clouds on the simulated radiances, and the method for accounting for clouds in the sounding retrieval process, is summarized as follows: (1) the OSSE vertical profiles of the cloud parameters, ice water content (CLDICE), liquid water content (CLDLIQ) and cloud fraction (CLOUD), along with the total column cloud amount (CLDTOT), are used to specify the PCRTM input cloud parameters consisting of optical thickness, effective cloud particle radius, and phase for each of the 18 possible CCSM3 model cloud levels ranging between 100.51 hPa and 992.56 hPa. The parameter conversion was accomplished assuming a cloud pressure thickness of 150 hPa and using empirical formulae relating the effective particle radius to optical depth for various types of cloud (e.g., those provided by Heymsfeld et. al. [33] for ice clouds). The cloud optical thickness was varied between 0.01 and 10, and a particle radius was then provided by the empirical relationships between cloud optical thickness and effective cloud particle radius, with a 10% random perturbation added to account for the cloud property measurement scatter about these relationships. PCRTM was then used to produce 19 radiance spectra (i.e., one for each cloud level and a clear-sky radiance spectrum). The DR retrieval is then performed from each of the 19 radiance spectra, providing 19 DR retrievals per grid point per month. It is important to note that the cloud height is assumed to be unknown for each DR retrieval, which is consistent with the DR application to satellite radiance observations. The monthly average retrieval for each model grid point that would have been obtained for each atmospheric level, assuming that the cloud radiances were observed from satellite observations throughout the entire month, is then approximated as a linear combination of the individual 19 profile retrievals weighted by the cloud fraction for each atmospheric level. That is:*T*(*p*) = [1 − Σ *cf_i_*(*p*)] **T*_0_(*p*) + Σ *cf_i_*(*p*) **T_i_*(*p*)   *i* = 1, 2,⋯, 18
(1) 
where *T*_0_ is the clear radiance retrieval value and *T_i_* is the cloud radiance retrieval value, and *cf_i_* is the cloud fraction, for each cloud level, *p*. The individual level cloud fractions are scaled so that their vertical total for each month and grid point is equal to the monthly mean value of the total cloud fraction, CLDTOT. It is noted that the linear combination assumption assumes that the satellite instrument is of relatively high spatial resolution, so that the impact of the horizontal distribution of multilayer cloud within a particular instrument FOV is minimal in the specification of the FOV cloud fraction for each cloud level. It is also important to note that since the same monthly mean model grid point atmospheric profiles are used to simulate the satellite radiance spectra, it is assumed that the atmospheric profiles being retrieved during the entire month for each model grid point geographical area are uncorrelated with the cloud properties associated with satellite observations corresponding to the true atmospheric profiles at each satellite FOV location and observation time.

## 6. Results and Discussion

Figure 1 shows a comparison between the “true” (i.e., the CCSM3 model forecast value) and the retrieved CO_2_ atmospheric mixing ratio variations in the 100-year OSSE. It can be seen that the DR retrieval algorithm is able to separate out the influence of CO_2_ variability from the atmospheric temperature variability influencing the variability in the radiance spectra. This separation shown for both the dependent and independent regression training data sets results from using radiances, for deriving the regression predictors, which cover a wide spectral range. It is particularly noteworthy that for the independent sample regression training, the CO_2_ concentration was assigned to each profile sample using a random number generator for a uniform distribution of values ranging over the entire range of values produced by CCSM3 for the 100-year period, without any reference to the time during the simulation period (i.e., time is not used as a regression predictor). Thus, the regression-predicted CO_2_ concentration is derived solely from the simulated IASI radiance spectrum, indicating a high degree of sensitivity of the radiance spectral features and magnitudes to the CO_2_ concentration value. This sensitivity is believed to be due to the Planck radiance temperature dependence, which increases greatly with decreasing wavelength, and the fact that CO_2_ absorption occurring at both short wavelengths, near 4.3 µm, and long wavelengths, near 15 µm, acts to separate the dependence of radiance variability due to both CO_2_ concentration variability from that due to atmospheric temperature variability. Otherwise, if measurements were used from only one of these spectral regions (i.e., the longwave or shortwave band), an error in the retrieval would occur, due to the ambiguity of the source of the atmospheric radiance variability (i.e., temperature or CO_2_ concentration variability).

Simulated retrievals from this 100-year climate model indicate that the decadal trends of atmospheric temperature and water vapor can be sensed with very useful accuracy under all-sky conditions (i.e., retrieved decadal trends are similar to the “Truth” defined as the 100-year climate model profile decadal trends). Figure 2 and Figure 3 show decadal trends retrieved for both the “clear-sky” and “all-sky” (i.e., cloud effects being included in the simulation) conditions. As can be seen, the “clear-sky dependent” sample trend retrievals nearly reproduce the “true” trend values, demonstrating the expected high accuracy obtained by using a linear regression algorithm for retrieving long period climate trends under clear conditions with representative statistics. However, it should also be noted that the differences in the accuracy of the “independent” and “dependent” sounding sample training indicate a significant sensitivity to the representativeness of the atmospheric profile data set used to derive the regression retrieval coefficients. Suggestions for addressing this dependence are addressed below.

Maybe the most striking retrieval result here is the agreement of “All-sky” and “Clear-sky” trends. While Dual Regression retrieval has demonstrated excellent capability for handling the challenge posed by clouds in the infrared, the agreement between the results obtained using the “all-sky” independent sample regression relations to those obtained using the “clear-sky” independent sample regression relations are nearly identical. The degree of insensitivity to the cloud conditions shown is primarily due to the fact that the DR retrieval algorithm enables a linear solution to the all-sky profile retrieval problem through the classification of the all-sky training sample with respect to cloud height [16]. However, the high degree of equivalence between the “clear-sky” and “all-sky” results obtained here might be enforced by the assumption that the cloud properties are statistically independent of the atmospheric profile conditions (which had to be made in this study in order to use the monthly mean CCSM3 output to simulate the individual cloudy sky radiances).

Figure 4 and Figure 5 show the regional 100-year trends in temperature and humidity, respectively. One can see that the geographical distribution of the temperature and moisture trends are retrieved from the simulated radiance data, with the possible exception of places with a relatively steep terrain (e.g., Greenland). The retrieval results obtained using the “dependent” training database are not shown, because they are virtually identical to the “truth”, as was also shown in Figure 2 and Figure 3. Although it can be seen that clouds introduce white noise within these regional trend patterns, especially for temperature, it is shown in Figure 6 that because this noise is spatially random, it tends to average out when producing the global mean trend values. It is also seen from Figure 6 that there is a relatively large vertically biased training sample dependence upon the retrieved moisture profile trends that does not result for the retrieved temperature profile trends. This may be a consequence of the nonlinear dependence of the variations of radiance with respect to the variations in specific humidity not completely accounted for by the cloud height classification process.

These results demonstrate a new level of effectiveness in decadal change atmospheric profile trend detection. As noted earlier, the same retrieval methodology has been applied to real AIRS, IASI and CrIS hyperspectral observations (Smith, N. et al. [20]) for obtaining atmospheric decadal trends. However, these data trends have been shown to be satellite instrument dependent, indicating that their absolute measurement accuracy is not good enough to infer climate trends over the relatively short time periods of their measurement history.

## 7. Summary and Conclusions

This study indicates that climate quality long-term decadal temperature and moisture profile trends can be retrieved using linear regression applied to CLARREO cross-calibrated operational satellite hyper-spectral infrared sounding radiance data. It is shown that the “all-sky” retrieval results are similar in accuracy to “clear-sky” results (see Figure 3). This result was expected because of the previously shown ability of the DR algorithm to deal with clouds through cloud-height classification of the all-sky regression training data sets [16]. However, it is also shown that there is a significant dependence of the retrieval accuracy on the representativeness of the profile/radiance training data set used to compute the regression retrieval coefficients. This result indicates that the training data set should be as representative as possible of the atmospheric conditions to be retrieved. One approach would be to use profiles and cloud conditions associated with reanalysis data sets, which are constantly updated by operational numerical weather prediction centers such as NCEP and ECMWF. Finally, it was also demonstrated, but not explicitly shown here, that multi-level clouds result in uncertainties similar to those for single level clouds, and have little added impact on retrieved globally averaged decadal trends.

These results support the conclusion that, with sufficient radiometric stability, accuracy and observational duration, it is possible to estimate trends in atmospheric temperature and humidity profiles over decadal to centennial time-scales. The immediate corollary to this is that such observations can and should be implemented in order to begin establishing such trends. Instruments such as CLARREO are specifically designed to achieve these goals, and should be implemented as soon as possible in order to establish a benchmark for the continued satellite observation of decadal trends. The observation of trends is essential for validating and improving climate models and their predictions that will enable humankind to both mitigate and prepare for environmental changes that impact the quality of life on Earth.

## Figures and Tables

**Figure 1 sensors-20-01247-f001:**
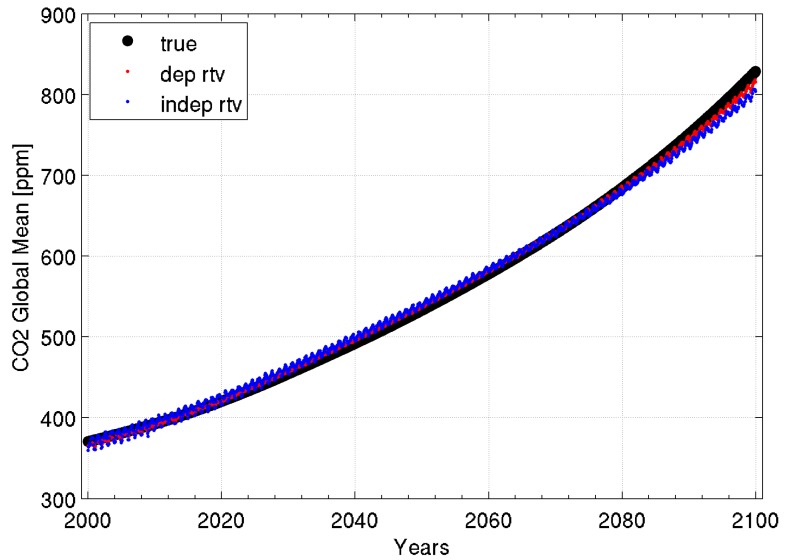
Comparison between the model global and monthly mean CO_2_ concentration (black dots) with two types of global and monthly mean CO_2_ concentration retrievals, derived from a dependent (red dots) and independent (blue dots) regression training set.

**Figure 2 sensors-20-01247-f002:**
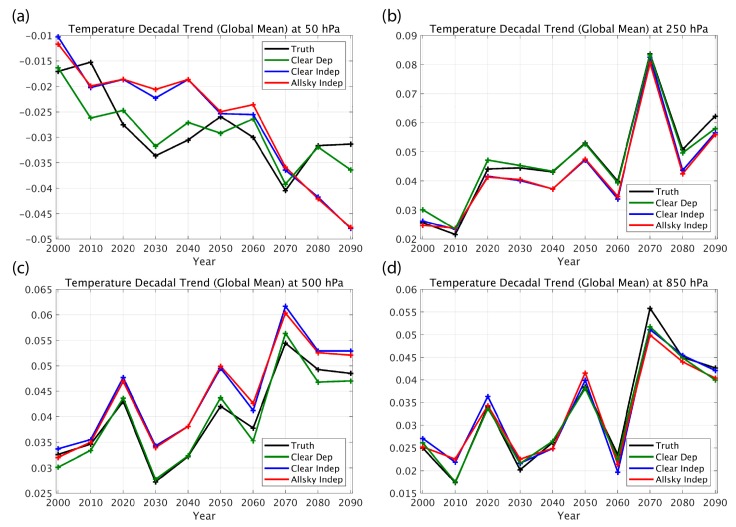
Examples of comparisons between decadal trends (K/year) produced by a 100-year climate model simulation for four atmospheric levels: (**a**) 50 hPa, (**b**) 250 hPa, (**c**) 500 hPa, (**d**) 850 hPa. The “Truth” (black curve) decadal trends throughout the twenty-first century are compared with those retrieved from simulated hyperspectral infrared radiance measurements, assuming: clear-sky conditions using a linear regression model trained with the climate model-produced atmospheric conditions (Clear Dep, green curve); clear-sky conditions using a linear regression model trained with contemporary global weather observation atmospheric conditions (Clear Indep, blue curve); and all-sky conditions using a linear regression model trained with contemporary global weather observation atmospheric conditions (All-sky Indep, red curve).

**Figure 3 sensors-20-01247-f003:**
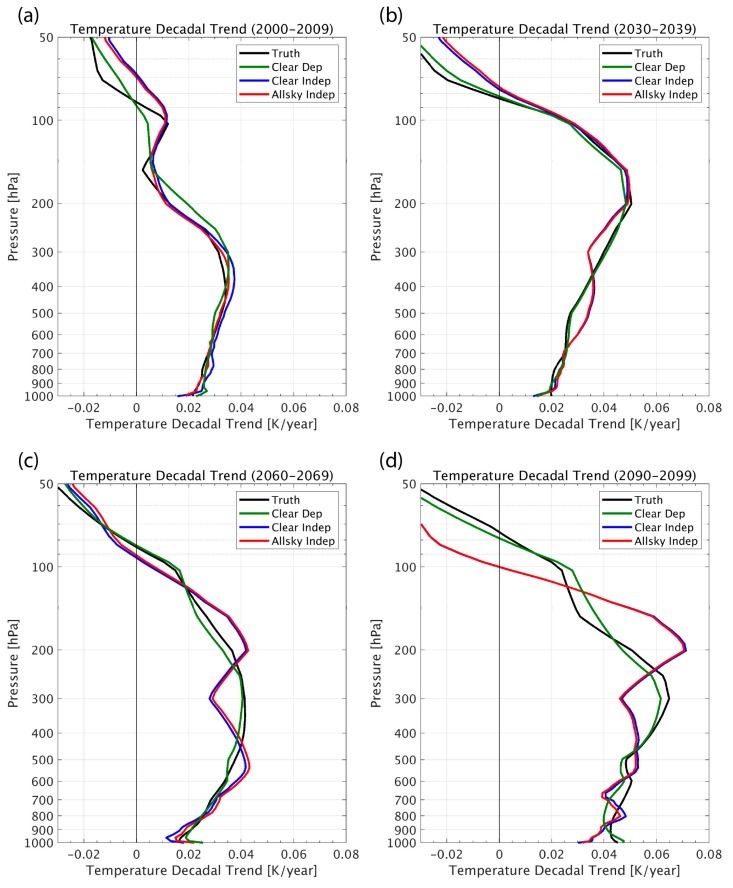
Examples of decadal trend profiles (K/year) produced by a 100-year climate model simulation for four decades: (**a**) 2000 - 2009, (**b**) 2030 - 2039, (**c**) 2060 - 2069, and (**d**) 2090 – 2099. The “Truth” (black curve) decadal trends are compared to decadal trends retrieved from simulated hyperspectral infrared radiance measurements, assuming: clear-sky conditions using a linear regression model trained with the climate model-produced atmospheric conditions (Clear Dep, green curve); clear-sky conditions using a linear regression model trained with contemporary global weather observation atmospheric conditions (Clear Indep, blue curve); and all-sky conditions using a linear regression model trained with contemporary global weather observation atmospheric conditions (All-sky Indep, red curve).

**Figure 4 sensors-20-01247-f004:**
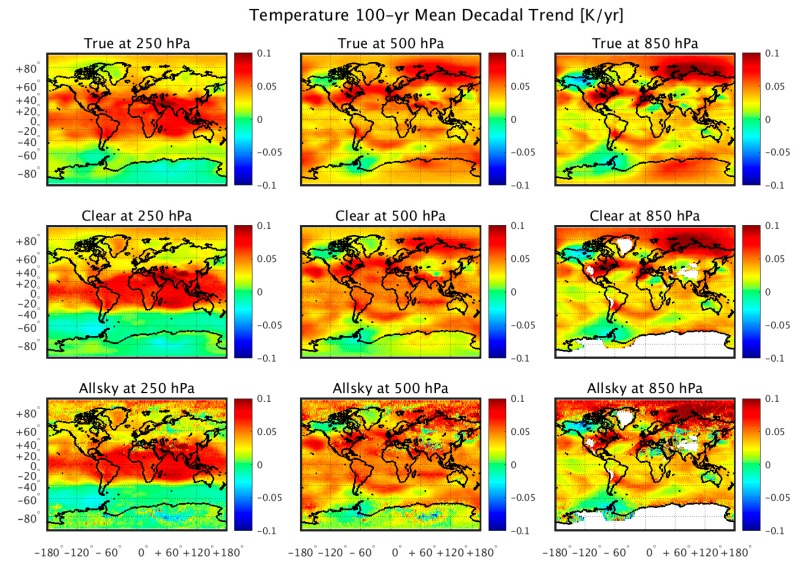
Examples of comparisons between the 100-year regional trends in atmospheric temperature (K/year) produced by a climate model simulation (top row) with 100-year trends throughout the twenty-first century retrieved from simulated CLARREO spectral infrared radiance measurements, assuming: clear-sky conditions using a linear regression model trained with contemporary global weather observation atmospheric conditions (middle row); and all-sky conditions using a linear regression model trained with contemporary global weather observation atmospheric conditions (bottom row). These results demonstrate the ability to retrieve regional temperature trend information when averaging over the full 100-year period.

**Figure 5 sensors-20-01247-f005:**
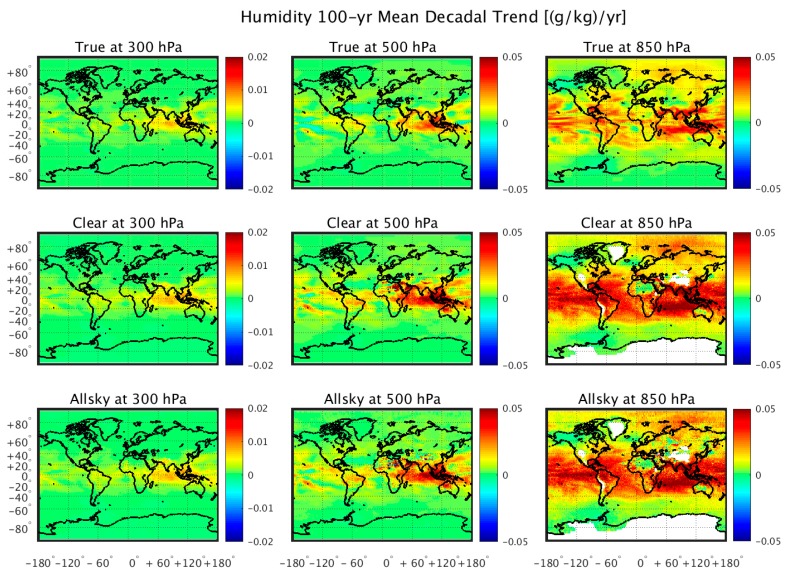
100-year regional trends of absolute humidity (g/kg/year) compared similarly to the temperature comparisons of Figure 4. These results demonstrate the ability to get useful regional information from the retrieval of atmospheric humidity.

**Figure 6 sensors-20-01247-f006:**
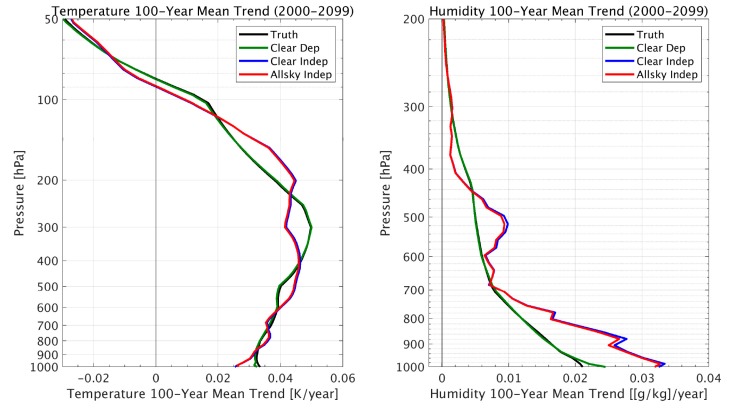
Comparison of retrieved global temperature (left) and humidity (right) trends with climate model observing system simulation experiment (OSSE) global trends for the 100-year period 2000–2099.

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
