# Peer review of "Retrieving Decadal Climate Change from Satellite Radiance Observations—A 100-year CO2 Doubling OSSE Demonstration"

_sensors, 2020, doi:10.3390/s20051247_

Round 1

Reviewer 1 Report

Review of “Retrieving decadal climate change from satellite radiance observations - a 100-year CO2 doubling OSSE demonstration”, by Smith Sr. and co-authors

(sensors-703340)

In this study, Smith Sr et al. use results of the CCSM3 OSSE to simulate IASI radiance spectra with the goal to determine whether reliable, unbiased decadal trends can be determined. 

Although it is clear that the study is well conceived and science is rigorous, there are some parts in the text that I found too condensed. Missing is a description of CLARREO’s spectral and horizontal resolutions compared to similar instruments already in orbit and a better description of the OSSE simulations and its purpose as an instrument emulator tool (in the present case, an emulator of CLARREO). I found myself going back and forth trying to figure out what was the place of CLARREO in this study, which was based on IASI’s measurements specifications. 

In the results, it could be really interesting and useful to show in more detail regional differences, for ex., plots as in Figure 3 but for Tropical/midlats/polar regions, and continental/oceanic areas. 

Minor corrections: please check lines 116/117, 334, 347/348; some of the references have are wrongly placed (e.g. [10] for Hilton et al. in line 117) and some have missing information (volume, doi, etc.).

Reviewer 2 Report

This is a very good paper.The authors illustrate how well of the IR sounders could be used in detecting the climate trend if they are in good calibration status. I have no comments. 

Author Response

Manuscript Accepted as submitted.  No response required.

Reviewer 3 Report

The submitted manuscript aims to determine whether CLARREO, as a mission, would be able to detect decadal-scale changes in atmospheric temperature and water vapor in an all-sky environment. As I understand it, CLARREO was designed to be a benchmark specifically to allow this, and a better question has already been asked and answered: how good does CLARREO need to be in order to serve as a benchmark for decadal-and-longer-term climate change. See https://doi.org/10.1175/JCLI-D-16-0704.1. (This and other papers really should have been cited; I found the reference list to be out of date. Also, the format of the references changed partway through the paper, from [##] to Author, (year).)

With the minimum benchmark already stated, and previous OSSE’s performed, I do not see the need for the present paper, except as a reminder that CLARREO is an excellent proposal for a mission.

Furthermore, while the authors made sure to state that their OSSE provides an “Upper bound” on the quality of CLARREO, the conclusions are stated as if this upper bound were a best estimate—with no statistics or error information to back up the claim. The authors acknowledge several sources of possible uncertainty, but I did not see any depiction of uncertainty in the results, including the figures.

The figures didn’t do much to support the main claim of the paper, either. Mostly, the pictures verified the idea of temperature and moisture retrieval from satellites…but even then, without statistics.

The paper is well-organized, but the writing and language need work. Some sentences are merely fragments (117–118). Some sentences are ponderous (see the sentence on 153–160). There were several sentences with missing articles, or with nouns and adjectives combined in chains.

Round 2

Reviewer 3 Report

I appreciate the authors' positive response to my earlier review, and the clarification about the objectives and methods of this paper. At this point, my remaining suggestion is to use a different color scheme for the line plots; lime green doesn't reproduce well in print, and green and red can be indistinguishable for the color blind.

Author Response

Figures were changed as suggested by the reviewer.